# Surgical Treatment and Complications of Deep-Seated Nodular Plexiform Neurofibromas Associated with Neurofibromatosis Type 1

**DOI:** 10.3390/jcm11195695

**Published:** 2022-09-26

**Authors:** Kunihiro Ikuta, Yoshihiro Nishida, Tomohisa Sakai, Hiroshi Koike, Kan Ito, Hiroshi Urakawa, Shiro Imagama

**Affiliations:** 1Department of Orthopedic Surgery, Nagoya University Graduate School of Medicine, 65 Tsurumai, Showa, Nagoya 466-8550, Japan; 2Department of Rehabilitation, Nagoya University Hospital, 65 Tsurumai, Showa, Nagoya 466-8550, Japan; 3Rare Cancer Center, Nagoya University Hospital, 65 Tsurumai, Showa, Nagoya 466-8550, Japan; 4Advanced Medicine, Nagoya University Hospital, 65 Tsurumai, Showa, Nagoya 466-8550, Japan

**Keywords:** neurofibromatosis type 1, nodular plexiform neurofibroma, enucleation, neurological deficits

## Abstract

Background: Nodular plexiform neurofibromas in individuals with neurofibromatosis type 1 often cause significant symptoms and are treated with surgical excision despite the potential risk of complications. This study aimed to clarify the surgical outcomes of deep-seated nodular plexiform neurofibromas and identify the factors associated with postoperative complications. Methods: We retrospectively reviewed patients with neurofibromatosis type 1 who underwent surgical excision for deep-seated nodular plexiform neurofibromas in our hospital from 2015 to 2021. Enucleation while preserving the nerve fascicles was attempted first, and en bloc resection, ligating the nerve origin in cases in which the parent nerve was entrapped by the tumor, making the tumor difficult to dissect, was performed. Results: In 15 patients, 24 nodular plexiform neurofibromas received surgical excision. Sixteen tumors were enucleated, and eight were en bloc resected. The symptoms of all 10 patients with preoperative symptoms resolved after surgery. Four patients developed new neurological deficits immediately after surgery, two of whom had retained neurological symptoms at the last visit, but these symptoms were mild. Conclusions: The present study demonstrates that surgical treatment of nodular plexiform neurofibromas, even deep-seated neurofibromas, is safe with a low risk of severe complications and improvement in preoperative symptoms.

## 1. Introduction

Neurofibromatosis type 1 (NF1) is an autosomal dominant inherited disorder caused by a germline mutation in the NF1 tumor-suppressor gene. Neurofibromas are a hallmark feature of NF1 and are divided into subtypes, including cutaneous, subcutaneous, and plexiform neurofibromas. Plexiform neurofibromas (PN) occur in approximately 30–50% of individuals with NF1 and may involve major peripheral nerves [1,2]. The majority of PN progress primarily during childhood [2,3]. PN can cause severe morbidity, including substantial pain, disfigurement, and neurological deficits; it has the potential of causing malignant transformation [4,5]. Although surgical treatment is indicated for symptomatic PN, its complete removal is frequently challenging due to significant risks of bleeding and neurological damage, especially in deep-seated tumors involving multiple nerves. In the United States and Europe, medical treatment using an MEK inhibitor, selumetinib, is currently available for pediatric patients with unresectable PN. Selumetinib has shown promising efficacy in inoperable PN and can be a feasible alternative to surgical resection for PN associated with major nerves [4]. However, this medication is not available for adult patients, and the role of surgical treatment remains significant in patients with NF1 who have the PN subtype.

PN may be diffuse or nodular and well-demarcated. The definition of nodular PN is mainly based on imaging findings. Diffuse PN spread extensively along connective tissue and surround normal structures with indistinct borders, and nodular PN form firm and round tumors and often present with multiple discrete tumors arising from peripheral nerves [6,7,8]. Several surgical techniques have been proposed for diffuse PN to decrease intraoperative bleeding and facilitate tumor excision [9,10,11,12,13]. These include electrosurgical procedures, adhesive or thrombogenic substances, or intravascular embolization prior to surgery. On the other hand, nodular PN compared with diffuse PN are likely to cause pain [8,14].

Recently, the concept of distinct nodular lesions (DNL) has been proposed to describe neurofibromas with a characteristic appearance on magnetic resonance imaging (MRI) [15,16]. DNL are round and well-demarcated, and their longest diameter is more than 3 cm with loss of central core signal, which is a characteristic of classic PN. DNL with rapid growth are of major concern because they can transform to atypical neurofibromatous neoplasms of uncertain biologic potential (ANNUBP) [16]. A biopsy to confirm the histologic diagnosis of PN (benign) is warranted in tumors with clinical or imaging findings that are suggestive of atypical behavior or malignant transformation.

Nodular PN that occur superficial to the fascia are almost exclusive of any sensory nerve origin and, thus, patients develop sensory paresthesia after surgical excision. However, excision of nodular PN that occur in the regions deeper than the fascia can cause motor paresis of the affected area. The optimal surgical technique for and frequency of postoperative complications after nodular PN resection are undetermined because few studies have investigated the surgical outcomes of nodular PN. 

In our hospital, more than 300 patients with NF1 have been followed up and treated by a multidisciplinary team [17]. Clinical management of deep-seated neurofibromas consists of monitoring disease progression and their malignant potential with MRI and treating the symptoms, including surgical excision of the tumors. Therefore, understanding the surgical outcomes of nodular PN is essential for patient management. The objective of the present study was to assess the surgical outcomes after excision of deep-seated nodular PN and to identify the factors related to complications in individuals with NF1 to inform surgical management.

## 2. Patients and Methods

### 2.1. Patients 

We retrospectively reviewed deep-seated nodular PN, which were surgically treated with intention to cure, at our institution from December 2015 to October 2021. This study was approved by the review board of our institution. All patients met the diagnostic criteria for NF1. The clinical data of the patients, including patient demographics, tumor characteristics (location, parent nerve, and size), surgical methods (enucleation and en bloc resection), histopathological diagnosis, and treatment outcomes were reviewed. We defined tumors located deep within myofascial compartments as deep-seated tumors. Tumors located superficial to the fascia and paraspinal area were excluded from the study. Owing to anatomical constraints, patients with nodular PN located in the thoracic and abdominal cavities were also excluded from this analysis. The pre-and postoperative symptoms of all patients were assessed. 

### 2.2. Diagnosis

In our institution, we routinely evaluated the presence of deep-seated tumors using whole-body MRI when patients consulted us. The preoperative MRI was available for review and evaluated in all patients. Nodular PN was defined as a solitary and well-demarcated mass, which does not involve diffuse PN based on MRI. The preoperative diagnosis of neurofibroma was confirmed histologically by excisional biopsy in three tumors and CT-guided core needle biopsy in two. In six tumors, the diagnosis of neurofibroma was made by an intraoperative frozen section. The remaining 13 tumors were preoperatively determined to be neurofibroma based on MRI or ^18^F-fluorodeoxyglucose positron emission tomography findings [18,19,20]. Postoperatively, all resected specimens were histologically analyzed and confirmed as neurofibroma by experienced pathologists through routine hematoxylin and eosin staining and immunohistochemical studies. Tumors diagnosed as ANNUBP according to the criteria by Miettinen et al. were excluded from the study [21].

### 2.3. Surgical Procedure

As patients with NF1 can have a number of neurofibromas throughout the body, we performed surgical excision on symptomatic or growing lesions. We attempted a macroscopically complete excision of the tumor entity. Enucleation (fascicle-preserving excision) was performed when the fascicles proximal and distal to the tumor could be identified and circumferentially dissected around the tumor. The tumor was incised using a nerve stimulator and/or microscopy to identify areas that did not contain motor fibers. When the nerve origin was entrapped by the tumor and the tumor was difficult to dissect, areas proximal and distal to the tumor were ligated and the tumor was removed (en bloc resection) (Figure 1). All procedures were performed by experienced orthopedic surgeons (K.I. (Kunihiro Ikuta), Y.N.). 

### 2.4. Statistical Analysis

The follow-up period was calculated from the date of the surgery to that of the last visit. We investigated whether tumor location (intramuscular or others), tumor size, the presence of preoperative biopsy, surgical methods (enucleation or en bloc resection), and types of parent nerve (major nerve or minor nerve) were associated with postoperative complications. In the upper extremity, the median, ulnar, radial, axillary, musculocutaneous, and brachial plexus nerves; in the lower extremity and pelvis, the femoral, sciatic, tibial, peroneal, obturator, and pelvic plexus nerves were defined as the major nerves according to a previous report [22]. Fisher’s exact tests were used for categorical data. Mann–Whitney U tests were used to compare medians of nonparametric continuous variables. *p* values < 0.05 were considered statistically significant. SPSS 27.0 for Windows software (SPSS, Inc., Chicago, IL, USA) was used for the statistical analyses.

## 3. Results

### 3.1. Patient and Tumor Characteristics

A total of 15 patients (eight females and seven males; mean age at surgery, 28.2 years [range, 13–43 years]) with 24 nodular PN located in the deep layer were surgically treated in our institution within the study period. All the nodular PN reviewed in this study were discrete and had no diffuse parts. Seven had a family history of NF1. The follow-up period ranged from 14 to 88 months (mean, 53.4 months). The tumors were located in the buttock (*n* = 5), upper arm (*n* = 2), thigh (*n* = 7), lower leg (*n* = 3), back (*n* = 4), neck (*n* = 1), and retroperitoneal space (*n* = 2). Fourteen tumors were located within the muscle and involved intramuscular nerves. Eight of the remaining 10 were intermuscular regions, and two were located in the retroperitoneum. Six of the 24 nodular PN involved major peripheral nerves (two in the sciatic nerve, one in the tibial nerve, two in the femoral nerve, and one in the musculocutaneous nerve). The median largest diameter at surgery was 5.8 cm (range, 1.0–11.4 cm). Patient and tumor characteristics are described in Table 1 and Table 2, respectively.

### 3.2. Surgical Outcomes

Of the 15 patients, two had multiple discrete lesions treated with different surgeries. Two patients received concurrent excision for multiple nodular PN. One surgery was performed to excise four discrete buttock tumors and another was performed to excise five discrete tumors of the thigh. Sixteen tumors were enucleated while preserving the nerve fascicles, and eight were en bloc resected. Of six nodular PN that originated from major nerves, five were enucleated, and one was en bloc resected (Figure 2). Enucleation was performed for eight of 14 intramuscular tumors, seven of eight intermuscular tumors, and one of two retroperitoneal tumors. Excluding two cases that underwent surgery for multiple tumors, the median operation time was 68 min (range, 10–267 min), and the median intraoperative blood loss was 28 mL (range, 1–385 mL). For extremity lesions, tourniquets were not used. Of the 15 patients, 10 presented with tumor-related symptoms before surgery. The chief complaints were pain in eight patients, numbness in one, and muscle weakness in one. The remaining five were scheduled for surgical excision due to growing lesions. The mean duration of symptoms prior to surgery was 9.8 months (range, two–26 months).

The preoperative pain symptoms reported by eight patients were resolved after surgery. Two patients with numbness or weakness also had improved symptoms after surgery. Complications occurred in five of 24 tumors (21%). Four patients developed new neurological deficits, but two recovered completely within a few days after surgery. Of the remaining two, one patient who underwent enucleation of a tumor that involved the saphenous nerve in the distal thigh developed hypoesthesia at the medial knee. The other had slight muscle weakness of the quadriceps and hypoesthesia at the anterior thigh after en bloc resection of the tumor in the retroperitoneum that involved the femoral nerve. These neurological deficits were retained at the last follow-up in both patients. Postoperative hemorrhage occurred in a patient who underwent enucleation of a nodular PN located in the musculocutaneous nerve. The hemorrhage was managed by pressing on the wound with a bandage. 

### 3.3. Factors Associated with Postoperative Complications

No significant relationships were detected between the occurrence of any complications and tumor location (*p* = 0.12), tumor size (*p* = 0.30), the presence of preoperative biopsy (*p* = 0.27), and surgical methods (*p* = 1.0). The occurrence of postoperative complications and the types of parent nerve showed a trend toward correlation (*p* = 0.078). Furthermore, we did not identify any factors associated with postoperative complications when the analysis was limited to neurological complications (Table 3).

## 4. Discussion

The present study showed that nodular PN could be resected without severe morbidities, even when major peripheral nerves in the deep layer are involved. This retrospective analysis is the first to evaluate surgical outcomes that are exclusive for nodular PN in patients with NF1. Although nodular PN can have a significant impact on neurological status and quality of life, the possibility of neurological complications has led to a cautious approach to the surgical management of nodular PN. Our series demonstrated that five tumors (21%) resulted in a postoperative complication and persistent neurological deficits occurred in two (8%) of the 24 nodular PN. Nguyen et al. described the surgical treatment of 56 PN, of which 13% developed new complaints after surgery [23]. Parada et al. reported that persistent sequelae were observed postoperatively in 17 of 96 (18%) pediatric patients with NF1 who have PN [24]. Although we are aware that no other study has focused exclusively on nodular PN, the results of our study are consistent with the findings of these previous studies [23,24]. However, since they did not analyze the outcomes of nodular PN separately from those of diffuse PN, our results should be interpreted with caution in comparison with the previous studies.

Neurofibromas consist of various cells, including Schwann cells, perineural cells, mast cells, fibroblasts, and macrophages [25]. Neurofibromas usually grow along the nerve, are confined within its epineurium, and envelope the fascicles of the nerve [23,26]. A retrospective series of pediatric patients with PN showed that complete tumor resection was possible in only 15% of the cases [27]. Therefore, enucleation of nodular PN, even with meticulous dissection, has generally been considered to increase the risk of transient or persistent neurological damage. 

On the other hand, several authors have suggested that it is possible to treat neurofibromas with fascicle-preserving enucleation [14,28]. In a large study of peripheral nerve tumors, Kim et al. showed that neurofibromas could be approached surgically in a similar manner to that of schwannoma, although the degree of fascicular involvement was different [28]. They examined the surgical outcomes of 99 NF1-associated neurofibromas and reported that most neurofibromas were successfully enucleated without severe deficits, even when a major peripheral nerve was involved. Donner et al. noted that most neurofibromas could be resected with minimal neurological damage when the surgical intervention was performed between the fascicles and when the proximal and distal ends of the tumor were identified [14]. We also paid significant attention to dissection, avoiding damage to the fascicles. In our cohort, 16 nodular PN (67%) were treated with enucleation. Two of the patients who underwent enucleation had neurological deficits, indicating that the rate of neurological complications after enucleation was 12%. Of these two patients, one had complete recovery of the neurological symptoms within a few days, while the other had persistent deficits at the last visit. Our high success rate in removing tumors with minor surgical morbidity was consistent with previous studies [14,28].

However, in some cases, despite careful dissection, one or more fascicles were identified as passing through the tumor entity. In such cases, fascicles must be sacrificed at both poles of the tumor during resection. Some studies have suggested that cuts of affected nerve fascicles did not cause neurological deficits because the nerve fascicles usually lose their function [14]. We performed en bloc resection for eight nodular PN in this study. Two patients (25%) developed neurological deficits (transient, 1 and permanent, 1), but the remaining six (75%) had no complications, although the parent nerves were sacrificed. Postoperative neurological deficits were unrelated to surgical methods (enucleation or en bloc resection). This was because six of the eight nodular PN treated with en bloc resection involved the intramuscular nerves. In our series, en bloc resection of neurofibromas arising from the intramuscular nerve did not cause persistent neurological deficits. A study reported that intramuscular peripheral nerve tumors were associated with denervation of the affected muscle [29]. The authors of the study observed a change in muscle denervation on the MRIs of some schwannomas (33.3%) and all ancient schwannomas (100.0%). Although it is necessary to confirm denervation with electrical stimulation intraoperatively, this finding may provide useful information for clinicians to explain possible outcomes of en bloc resection to patients before the surgery. 

Preoperative biopsy may cause inflammation and scarring within the lesion, resulting in adhesion of the tumor to the fascicles. Neurofibromas circumferentially enclose the parent nerve, increasing the risk of injury during excision [30]. Perez-Roman et al. reported that preoperative biopsy was significantly associated with postoperative deficits and neuropathic pain. Their study performed percutaneous or open biopsies on 10 neurofibromas, with increased neurological deficits in five (50%) cases after surgery [31]. In the current study, preoperative biopsies were performed on five nodular PN, but postoperative neurological symptoms occurred in only one (20%) retroperitoneal case treated with en bloc resection. In our cohort, the presence or absence of a preoperative biopsy did not affect the occurrence of neurological deficits.

The main goal of surgical treatment is to improve PN-associated morbidity. The prevalence of PN-related pain in individuals with NF1 was reported to be 37–59% [23,32,33]. Half (53%) of the patients in the current series had pain as their primary symptom, and all were relieved from pain after surgery. Nguyen et al. reviewed the surgical outcomes of 52 patients with NF1 with 56 PN, including 23 nodular PN. They reported that 14 of 20 patients with pain as the primary symptom achieved surgical pain relief [23]. 

This study has several limitations. First, the study is limited by its retrospective design. Second, it includes a small number of patients, and some had a short follow-up period. Third, the surgical technique to preserve or not preserve fiber bundles is dependent of intraoperative findings and the decision of the surgeon (two different surgeons were included). Despite these limitations, to the best of our knowledge, this is the first case series to exclusively report surgical outcomes of deep-seated nodular PN.

## 5. Conclusions

The present study demonstrated that the surgical outcomes of deep-seated nodular PN were reasonably good. Furthermore, we could accomplish enucleation for most deep-seated nodular PN with acceptable risk to the parent nerves involved. Although clinical factors related to postoperative complications were not found, our results provide important information for surgeons who intend to attempt the surgical excision of symptomatic deep-seated nodular PN.

## Figures and Tables

**Figure 1 jcm-11-05695-f001:**
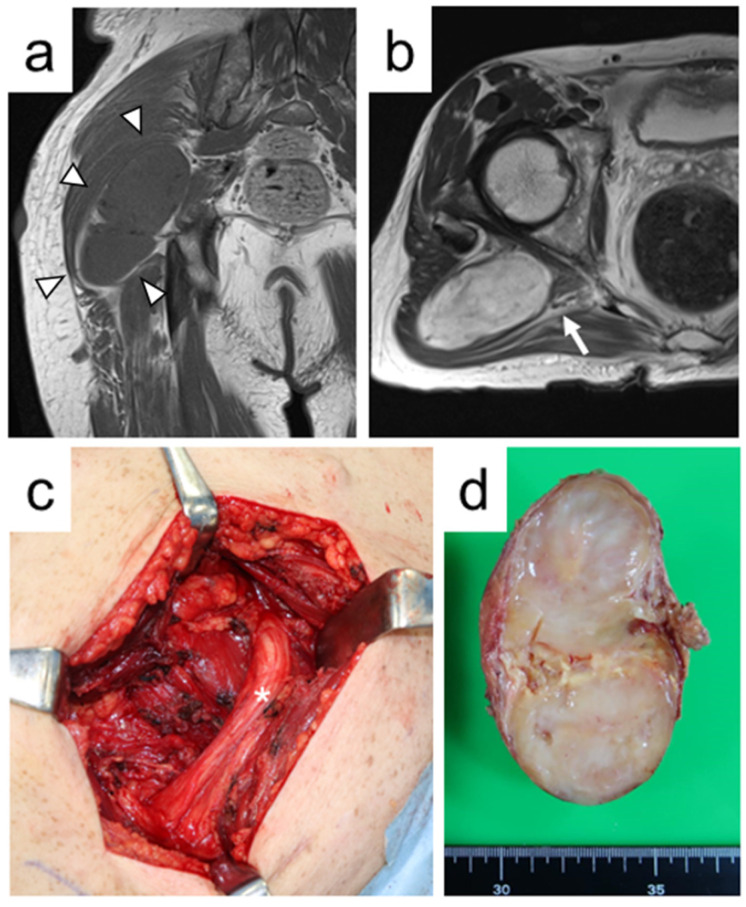
A 55-year-old male with nodular plexiform neurofibroma of the right buttock (Case 7). (**a**) Coronal T1-weighted magnetic resonance image shows an intramuscular tumor with homogeneous iso-signal intensity compared with skeletal muscle (arrowheads). (**b**) Axial T2-weighted magnetic resonance image reveals that the tumor between the piriformis muscle and gluteus maximus muscle is adjacent to the sciatic nerve (arrow). (**c**) An intraoperative photograph shows that the tumor originated from the branch of the sciatic nerve (asterisk) to the gluteus maximus muscle and was treated with en bloc resection. (**d**) The en bloc resected specimen reveals a yellowish-white tumor on gross finding (Tumor 10).

**Figure 2 jcm-11-05695-f002:**
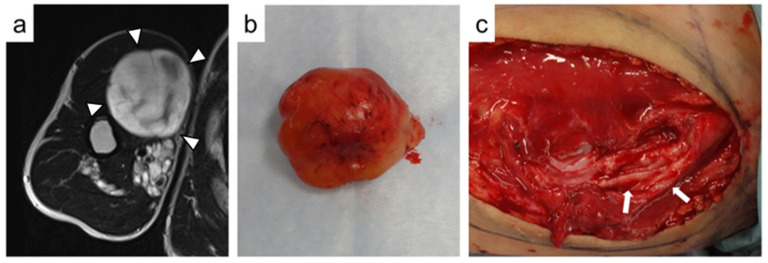
A 26-year-old male with nodular plexiform neurofibroma in the right upper arm (Case 8). (**a**) On axial T2-weighted magnetic resonance image, the tumor shows high signal intensity with some hypointense areas (arrowheads). (**b**) The tumor is successfully enucleated (Tumor 11). (**c**) An intraoperative photograph showed that the musculocutaneous nerve is preserved with a capsule after enucleation of the tumor (arrows).

**Table 1 jcm-11-05695-t001:** Clinical features of 15 patients with nodular plexiform neurofibroma.

Case No.	Age at Surgery (ys)	Gender	Number of Tumors	Indication for Surgery	Duration of Symptoms (ms)	Follow-Up Duration (ms)
1	38	F	1	numbness	26	44
2	32	F	1	pain	6	42
3	43	M	1	tumor growing	-	46
4	22	M	1	muscle weakness	3	69
5	29	F	1	pain	5	67
6	28	F	4	tumor growing	-	14
7	55	M	1	pain	2	43
8	26	M	1	pain	18	68
9	14	M	1	tumor growing	12	72
10	17	M	1	tumor growing	-	41
11	14	F	2	pain	8	52
12	39	F	6	pain	4	60
13	17	F	1	pain	19	51
14	12	M	1	pain	13	88
15	36	F	1	tumor growing	-	44

No. number, M male, F female, ys years, ms months.

**Table 2 jcm-11-05695-t002:** The detailed information of 24 nodular plexiform neurofibromas.

Tumor No.	Site	Location	Size (cm)	Surgical Methods	Postoperative Complications
1	back	intramuscular	6.1	en bloc resection	-
2	back	intramuscular	8.7	en bloc resection	-
3	back	intramuscular	4.0	en bloc resection	-
4	lower leg	intermuscular (tibial nerve)	5.0	enucleation	-
5	buttock	intermuscular (sciatic nerve)	5.8	enucleation	-
6	buttock	intermuscular	5.6	enucleation	-
7	buttock	intramuscular	5.0	enucleation	-
8	thigh	intermuscular (sciatic nerve)	5.8	enucleation	-
9	thigh	intramuscular	2.6	enucleation	-
10	buttock	intermuscular	6.8	en bloc resection	-
11	upper arm	intermuscular (musculocutaneous nerve)	8.0	enucleation	bleeding
12	lower leg	intramuscular	6.5	en bloc resection	sensory deficits (transient)
13	retroperitoneum	intermuscular (femoral nerve)	6.6	enucleation	muscle weakness (transient)
14	upper arm	intramuscular	1.9	enucleation	-
15	neck	intramuscular	0.8	enucleation	-
16	back	intramuscular	11.4	enucleation	-
17	thigh	intermuscular	5.1	enucleation	-
18	thigh	intramuscular	6.5	enucleation	-
19	thigh	intramuscular	7.3	enucleation	-
20	thigh	intramuscular	5.0	enucleation	-
21	thigh	intermuscular (saphenous nerve)	4.3	enucleation	sensory deficits (persistent)
22	buttock	intramuscular	4.3	en bloc resection	-
23	lower leg	intramuscular	10.6	en bloc resection	-
24	retroperitoneum	intermuscular (femoral nerve)	6.8	en bloc resection	muscle weakness and sensory deficits (persistent)

No. number.

**Table 3 jcm-11-05695-t003:** The association between clinical variables and postoperative neurological complications in 24 nodular plexiform neurofibromas.

Variables	Number of Tumors (*n* = 24)	Tumors without Neurological Complications (*n* = 20)	Tumors with Neurological Complications (*n* = 4)	*p* Value
Tumor size ^a^ (median, cm)	5.8	5.6	6.6	0.63
Tumor location				0.27
Intramuscular	14	13	1	
Others	10	7	3	
Presence of biopsy				1.0
Yes	5	4	1	
No	19	16	3	
Surgical methods				0.58
Enucleation	16	14	2	
En bloc resection	8	6	2	
Types of parent nerve				0.25
Major nerve	6	4	2	
Minor nerve	18	16	2	

^a^*p* value is calculated by Mann-Whitney U test.

## Data Availability

The datasets used and analyzed during this study are available from the corresponding author upon reasonable request.

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
