# Peer review of "Surgical Treatment and Complications of Deep-Seated Nodular Plexiform Neurofibromas Associated with Neurofibromatosis Type 1"

_jcm, 2022, doi:10.3390/jcm11195695_

Round 1

Reviewer 1 Report

A well designed, well written and well documented retrospective case series. Please consider the following points:

      1. I suggest that the Authors move the paragraph re: MRI diagnosis of PN from Discussion to Introduction.

     2.   Line 75: I suggest to remove “24” as that datum belongs to – and is redundant with – the Results section.

     3. Line 206: “caution” not “cation”.

Author Response

Reviewer 1

A well designed, well written and well documented retrospective case series. Please consider the following points:

Thank you for your review of our paper. We have answered each of your points below. The changes in the manuscript are shown in red font.

  1. I suggest that the Authors move the paragraph re: MRI diagnosis of PN from Discussion to Introduction.

Response: Thank you for this suggestion. We agree with your suggestion. This paragraph has been moved to the Introduction (L60-67) and Results sections (L144-145) in the revised manuscript.

  1. Line 75: I suggest to remove “24” as that datum belongs to – and is redundant with – the Results section.

   Thank you for pointing this out to us. We have corrected the text accordingly (L84).

  1. Line 206: “caution” not “cation”.

   Thank you for pointing this out to us. I have corrected it (L218).

Reviewer 2 Report

Authors nicely describe a retrospective series of 24 patient with surgical treatment and complications of deep-seated nodular plexiform neurofibromas.

In general:
Although it is of interest what the complication profile is in this group, I would be interested in the recurrence ratio of NFs after surgery. Also, PROMs would be very useful in this group: do patients actually benefit from this type of surgery. However, in this retrospective cohort probably no PROMs can be obtained.

specific comments:
re line 99: were ANNUBPS and atypical NFs operated in same fashion as the  neurofibromas? why not include them in this study, as they are not easily differentiation before resection.

re line 106: what neuromonitoring was used?

re line 124: what is a major nerve? mixed nerve? please be more specific

re line 280: this might be the first paper on PN alone, however more papers have described the resection of PNs as part of more types of nerve tumors. Therefore, I think the claim is not correct. Please rephrase

Author Response

Reviewer 2

Authors nicely describe a retrospective series of 24 patient with surgical treatment and complications of deep-seated nodular plexiform neurofibromas.

In general:
Although it is of interest what the complication profile is in this group, I would be interested in the recurrence ratio of NFs after surgery. Also, PROMs would be very useful in this group: do patients actually benefit from this type of surgery. However, in this retrospective cohort probably no PROMs can be obtained.

Response: Thank you for your comment. We did not mention recurrence in the manuscript because we did not have a postoperative protocol for MRI to evaluate recurrence of nodular plexiform neurofibromas in this study. We performed whole-body MRI as a survey of NF1 at a frequency dependent on the patient, and had no recurrence during the follow-up period. Unfortunately, we do not have PROMs for NF1 patients at our institution, and we would like to do so in the future.

specific comments:
re line 99: were ANNUBPS and atypical NFs operated in same fashion as the neurofibromas? why not include them in this study, as they are not easily differentiation before resection.

Response: Thank you for your comment. For ANNUBPs and atypical NFs, we recently indicated en bloc marginal resection (ref. 26); therefore, they were excluded from the study. We excluded atypical NFs and ANNUBPs from this analysis to prioritize the uniformity of the treatment strategy and the uniformity of subjects. However, as you indicated, it is not easy to differentiate neurofibromas from atypical NFs before resection, and we have encountered cases in which the diagnosis of neurofibroma by biopsy turned into atypical NF in the surgical specimen.

re line 106: what neuromonitoring was used?

Response: Thank you for this suggestion. We used a nerve stimulator to identify the motor nerves. We have corrected this in the revised manuscript (L114).

re line 124: what is a major nerve? mixed nerve? please be more specific

Response: As you have indicated, the description of “a major nerve” was inadequate in the manuscript. The major nerves were classified based on previous reports (ref. 22). The following definitions have been added to the text (L133-136). In the upper extremity, the median, ulnar, radial, axillary, musculocutaneous, and brachial plexus nerves; in the lower extremity pelvis, the pelvic plexus, femoral, sciatic, tibial, peroneal, saphenous, sural, and obturator nerves were defined as the major nerves.

re line 280: this might be the first paper on PN alone, however more papers have described the resection of PNs as part of more types of nerve tumors. Therefore, I think the claim is not correct. Please rephrase

Response: Thank you for your comments. We think that our report is the first to focus solely on deep-seated nodular plexiform neurofibromas and to analyze their results. However, as you pointed out, other authors have previously reported the results for plexiform neurofibromas, including nodular plexiform neurofibromas (deep-seated). In this conclusion, we omitted the word "for the first time” (L204).

Reviewer 3 Report

Comment

Thanks the authors to submit a comprehensive study addressing “Surgical treatment and complications of deep-seated nodular plexiform neurofibromas associated with neurofibromatosis type 1.” This is an interesting topic of surgical treatment of nodular plexiform neurofibromas, even deep-seated neurofibromas with less complications. Here are some feedbacks for more information needed.

1.      I am wondering if the deep-seated nodular plexiform neurofibromas in your series possibly were close to the vessels or were fed by the vessels. Please address if the pre-operative embolization could be performed for those even bigger-size tumor prior to the surgical intervention.

Please refer to “Hsu, C. K., Denadai, R., Chang, C. S., Yao, C. F., Chen, Y. A., Chou, P. Y., ... & Chen, Y. R. (2022). The Number of Surgical Interventions and Specialists Involved in the Management of Patients with Neurofibromatosis Type I: A 25-Year Analysis. Journal of Personalized Medicine12(4), 558.

2.      For the deep seated NF1 in this study, there is no tumor located or invading to the thoracic cavity or intra-abdominal cavity. Are they excluded from your series, please address in detail in the materials and methods. By the way, what is the different of major nerve and minor nerve, please shortly define them in the article.

3.      As for more clear understanding easily for the readers. Could you please provide an algorithm to educate the readers the steps of decision-making based on your experiences, for examples: tumor sizes, intramuscular or intermuscular, enbloc or enucleation. Because the table 3 seems the most important finding related to the key of this study, a clear pathway of the summary of suggestion is needed.

Figure 1, please add the arrow or mark to the mass in (a). and (d) what is the number of tumor in table 2, please correlate and describe.

Figure 2, please show the mass with mark in (a). and (b) what is the number of tumor in table 2, please correlate and describe.

Pease take care of the above suggestions, then acceptance will be considered.

Author Response

Reviewer 3

Thanks the authors to submit a comprehensive study addressing “Surgical treatment and complications of deep-seated nodular plexiform neurofibromas associated with neurofibromatosis type 1.” This is an interesting topic of surgical treatment of nodular plexiform neurofibromas, even deep-seated neurofibromas with less complications. Here are some feedbacks for more information needed.

 We appreciate your comments. We have answered each of your points below.

  1. I am wondering if the deep-seated nodular plexiform neurofibromas in your series possibly were close to the vessels or were fed by the vessels. Please address if the pre-operative embolization could be performed for those even bigger-size tumor prior to the surgical intervention.

Please refer to “Hsu, C. K., Denadai, R., Chang, C. S., Yao, C. F., Chen, Y. A., Chou, P. Y., ... & Chen, Y. R. (2022). The Number of Surgical Interventions and Specialists Involved in the Management of Patients with Neurofibromatosis Type I: A 25-Year Analysis. Journal of Personalized Medicine, 12(4), 558.”

Response: Thank you for your comment. The developmental forms of nodular plexiform neurofibroma can be divided into those that occur alone and those that occur in combination with diffuse plexiform neurofibroma. In the former, none of the tumors had a lot of vascular supply that required embolization in our series. Therefore, we were able to perform enucleation or en bloc resection, with relatively little blood loss. On the other hand, diffuse plexiform neurofibromas, as shown in the literature you presented, are characterized by abundant blood flow. It is useful to excise them after embolization of the vessels feeding them. Thank you for helpful suggestion. We have cited a report by Hsu, C. K et al., in the introduction section as ref.13 in the revised manuscript (L55-56).

  1. For the deep seated NF1 in this study, there is no tumor located or invading to the thoracic cavity or intra-abdominal cavity. Are they excluded from your series, please address in detail in the materials and methods. By the way, what is the different of major nerve and minor nerve, please shortly define them in the article.

Response: Thank you for your indication. We also treated NF1 patients in our multidisciplinary team, so we often encounter tumors arising in the mediastinum or close to the abdominal organs. However, orthopedic surgeons do not perform surgery for thoracic or intra-abdominal lesions. Therefore, we have not included these non-specialty cases in our series and have added them to the METHODS section (L92-93).

We apologize for not defining the major nerve. The major nerves were classified based on previous reports (ref 22). The definitions have been added to the text (L133-136). Namely, the major nerves of the upper extremity are the median N, ulnar N, radial N, axillary N, musculocutaneous N, and brachial plexus; in the lower extremity pelvis, the pelvic plexus, femoral N, sciatic N, tibial N, peroneal N, saphenous N, sural N, and obturator N in the lower extremity pelvis. We have added these to the METHODS section (2.4).

  1. As for more clear understanding easily for the readers. Could you please provide an algorithm to educate the readers the steps of decision-making based on your experiences, for examples: tumor sizes, intramuscular or intermuscular, enbloc or enucleation. Because the table 3 seems the most important finding related to the key of this study, a clear pathway of the summary of suggestion is needed.

Response: Thank you for helpful suggestion to refine our paper. It would have been desirable if we could have extracted the clinical factors involved in the development of complications, but the number of subjects was also small; therefore, we need to accumulate more cases for further study. Basically, if we can identify the nerve fascicles of the tumor, enucleation is attempted in all the cases. In our experience, nodular plexiform neurofibromas arising from “unnamed” intramuscular nerves can be ligated by en bloc resection without complications. When denervation can be confirmed intraoperatively using a nerve stimulator, the nerve fascicles can be cut, and the patient may be asymptomatic. In cases where the tumor grows over time, other nerve networks may compensate for this function. However, because sensory deficits cannot be evaluated with a nerve stimulator, the primary approach is enucleation with the aim of preserving nerve fascicles. This may not be an answer to your question, but it is difficult at present to provide a clear algorithm, and we are concerned that this may be misleading.

Figure 1, please add the arrow or mark to the mass in (a). and (d) what is the number of tumor in table 2, please correlate and describe.

Response: We have added arrowheads showing the tumor within Figure 1a. This was the case for tumor No. 10. We have made correction in the description of text in the Figure legends (L120-127).

Figure 2, please show the mass with mark in (a). and (b) what is the number of tumor in table 2, please correlate and describe.

Response: Arrowhead indicating the tumor have been added in Figure 2a. This was the case for tumor No. 11. The description of text in the Figure legends have been changed (L176-178).

Pease take care of the above suggestions, then acceptance will be considered.

Round 2

Reviewer 3 Report

Dear authors,

The revised form seems adequate with full of knowledge for publication.